# The Psychosocial Burden of Families with Childhood Blood Cancer

**DOI:** 10.3390/ijerph19010599

**Published:** 2022-01-05

**Authors:** Florencia Borrescio-Higa, Nieves Valdés

**Affiliations:** Business School and GobLab UAI, Universidad Adolfo Ibáñez, Santiago 7941169, Chile; nieves.valdes@uai.cl

**Keywords:** childhood cancer, psychosocial cost, well-being, caregiver, siblings

## Abstract

Cancer is the second leading cause of death for children, and leukemias are the most common pediatric cancer diagnoses in Chile. Childhood cancer is a traumatic experience and is associated with distress, pain, and other negative experiences for patients and their families. Thus, psychosocial costs represent a large part of the overall burden of cancer. This study examines psychosocial experiences in a sample of 90 families of children with blood-related cancer in Chile. We provide a global overview of the family experience, focusing on patients, caregivers, and siblings. We find that most families report a negative impact upon diagnosis; disruptions in family dynamics; a range of negative feelings of the patient, such as depression, discouragement, and irritability; and difficulty with social lives. Additionally, they report negative effects in the relationship between the siblings of the patient and their parents, and within their caregivers’ spouse/partner relationship, as well as a worsening of the economic condition of the primary caregiver. Furthermore, over half of the families in the sample had to move due to diagnosis and/or treatment. Promoting interventions that can help patients, siblings, and parents cope with distress and promote resilience and well-being are important.

## 1. Introduction

Although cancer survival rates for children and adolescents have improved in recent decades, cancer is the second leading cause of death for children ages 5–15 in Chile [1]. At a global level, leukemias are the most common pediatric cancer diagnoses, and are more expensive to treat than other types of pediatric cancers [2,3]. The Chilean childhood cancer registry reports that between 2007 and 2011, there were 965 new cases of leukemia among children under the age of 15, with a rate of 51.5 per million [1].

The treatment of children with cancer is expensive and resource intensive worldwide, and Chile is no exception [4,5]. However, the medical cost is not the only cost for families; several other associated costs contribute to the overall burden of cancer, namely, direct (e.g., medical care), indirect (loss of resources and opportunities), and psychosocial [6,7,8,9]. The latter encompasses intangible costs associated with cancer, such as pain and suffering, and the additional cost to individuals’ well-being [8].

A cancer diagnosis and its treatment can have a substantial impact not only on the patient, but on the daily lives of all family members, as it can cause psychological distress, pain, and other negative experiences. Most families of children with cancer experience significant distress throughout diagnosis and treatment [10]. Not only the child with cancer experiences significant psychological distress, but family members often present a series of behavioral changes or symptoms of anxiety and depression [11].

In particular, post-traumatic stress symptoms have been documented in parents of children with cancer [12,13,14]. A review article finds that parents experience psychological stress, mostly around the time of diagnosis, with mothers reporting more symptoms than fathers [15]. Parental distress can have a negative impact on quality of life, family functioning, and marital distress, and may have an impact on the well-being, coping, and adjustment of the diagnosed child and other children [16,17].

In a systematic review, Alderfer et al. find that siblings of children with cancer often experience post-traumatic stress symptoms, negative emotional reactions, and poor-quality family and social lives [18]. Based on a survey of 125 siblings, another study finds that siblings of children with cancer present cancer-related post-traumatic stress reactions [19].

Most of the above-mentioned studies are based on evidence from developed countries. This is consistent with a recent systematic review by Essue et al., who find that most studies are focused on the financial impact of cancer, and are based on adult populations, or are conducted in developed countries [8].

The purpose of this study is to understand the psychosocial costs of blood cancer for pediatric patients and their families, with the goal of creating a resource for policymakers to improve programs that provide childhood cancer services. These programs should not only enhance financial coverage but also provide psychological support for children and other family members. We surveyed caregivers of children with blood cancer in Chile to analyze the psychosocial experience of families with a child with blood cancer.

## 2. Methods

### 2.1. Data/Survey

We designed a 20-minute phone survey centered around the social and emotional impact of a blood-related cancer diagnosis on the child and their family. The survey design was a collaborative work with researchers at UAI Metrics in the School of Psychology at Universidad Adolfo Ibáñez, who also implemented the survey. Participants were recruited in three ways: (1) through the parent liaisons of the *oncomamas* group, that gathers mothers of children with cancer in Chile, (2) through a social media campaign, and (3) using a snowball sampling technique. Interested participants received information regarding the study and the link to the online informed consent form. Those who signed the consent form were then contacted by phone to conduct the survey. The survey was anonymous and voluntary. This study was approved by the Universidad Adolfo Ibañez Institutional Review Board (IRB Protocol 03/2020).

The inclusion criterion for the survey study was being a parent or guardian of a child with a diagnosis of a blood-related cancer before the age of 18. We focused on two of the most frequent categories of cancer diagnosis for children: leukemia and non-Hodgkin’s lymphoma [1].

A total of 106 participants answered the survey, which was performed in two waves: January–April 2020 and July–October 2020. We excluded 16 participants as their diagnoses did not correspond to a blood-related cancer. Therefore, the final analysis included 90 survey responses.

The survey was organized into modules that focused on the socio-demographic information of the child (age, gender, region of birth, type of health insurance, age at diagnosis, diagnosis, treatment status and household composition) and the primary caregiver (age, socio-economic status and occupation). Furthermore, survey questions were centered on the impact on the family: whether they had to move to a new city due to the treatment; the educational, social, and emotional impacts on the patient and their siblings; and a description of the main activities of the primary caregiver, their labor market outcomes, and the impact on the parents’ spouse/couple relationship.

The survey did not include explicit questions on the patients’ socio-economic status in order to concentrate on non-economic costs and avoid missing values in relevant variables. Consequently, we constructed a proxy for socio-economic status (SES) using information on the region of residence and health insurance affiliation, as evidence suggests that both variables are highly correlated with socio-economic status in Chile [20,21,22]. Private health insurance is prevalent among high-income households. Accordingly, we considered all children with private health insurance as high SES. Public insurance is divided into four tiers that have increasing copayments associated with higher household income. Thus, we classified all publicly insured children covered that belong to the top tier, with the highest per-capita income level and highest copayments, as high SES. The classification of children in the second and third tiers was not straightforward as the purchasing power of families with relatively similar income levels may differ according to the region of residence. Thus, we classified all children in the second and third tiers, who reside in a region with a higher average income level than the national average, as high SES. In summary, we defined SES as high if one of the following conditions applied: (1) the child is a beneficiary of private health insurance; (2) the patient is a beneficiary of public health insurance, and the family income level puts the family in the highest copayment tier; or (3) the child is publicly insured and belongs to the second or third copayment tier, and the region of residence has an average income level above the national average. All of the remaining cases were considered low SES.

A relevant dimension of the analysis is the region of birth, since Chile is a highly centralized country with almost 40% of the population living in the metropolitan region, where the capital of the country and most advanced health care institutions are located [23].

To assess representativeness, in the Appendix A section A, we compare the descriptive statistics of the sample with those obtained from the “First Report of the Chilean National Childhood Cancer Registry”, which collected information on children aged 14 years old or under, diagnosed with any type of cancer between 2007 and 2011 [1].

### 2.2. Statistical Methods and Empirical Model

Descriptive statistics of the sample (*n* = 90) are presented as percentages for categorical variables and averages for continuous variables. We focused on the impact of the disease on the following dimensions: (i) family overall, (ii) patient, (iii) siblings, and (iv) caregiver. For each of these dimensions, we present survey results as percentages of categorical variables. Supplementary Material B presents detailed descriptions of the survey questions used in the analyses.

We use spider graphs to present a depiction of the relationship between the impact of the disease on the patients, their siblings and the primary caregivers. These graphs simultaneously show the incidence of selected impacts of SES on the child. We categorized the outcomes into two groups: the impact on social life and emotions, and the overall impact on the family, as well as educational and occupational outcomes.

Finally, we analyzed the association between the selected impacts of the disease and the characteristics of the patient, the primary caregiver, and the family. All outcomes were measured as binary variables. Accordingly, we estimated the parameters of logit models by maximum likelihood, and we computed the marginal effects as the average of the individual effects. In all regressions, we included the following set of covariates: (1) an indicator of the child’s gender; (2) the age of the child in years; (3) an indicator variable that takes a value of 1 if the child was born in the metropolitan region; (4) an indicator variable that takes a value of 1 if the child lives with a single parent; (5) an indicator of high SES. To assess the magnitude of the marginal effects, we compared them with unconditional probabilities, computed as the number of observations that reported experiencing the specific impact, divided by the total number of observations considered in the regression. Standard errors used to calculate *p*-values are robust to the presence of heteroskedasticity.

## 3. Statistical Analysis

### 3.1. Characteristics of the Patients and Their Families

Table 1 presents the descriptive statistics of the sample. Gender was balanced in the survey, with 52.2% of observations for female patients. The age at diagnosis ranged from 0 to 16 years old, with an average value of 6.3. We categorized patients’ ages into brackets of 3 years and found a similar number of observations in each bracket. Approximately 40% of patients were born in the metropolitan region, consistent with the population distribution at the country level. The most prevalent diagnosis in the survey was leukemia, with 92% of observations. At the time of the survey, 66.7% of the patients were in treatment, 31.1% had finished it, and 2.2% had died. Patients were evenly covered by either private or public health insurance (47.8%), and only four children were covered by other insurance systems (military forces, police, or not specified). Most patients lived in households with both parents present (48.9%), 10% lived with both parents and other adults and 72.2% of patients had siblings. Considering the definition explained in the previous section, we classified 63.3% of the children as belonging to a high SES. For almost all patients, the primary caregiver was the mother (92.1%). Furthermore, in 67.4% of the observations, the father was one of the child’s caregivers. Slightly over half of the primary caregivers (50.6%) were workers (employers, employees or self-employed, working either part or full time), and 34.6% were homemakers.

### 3.2. Impact on the Family

Table 2 shows the overall impact on families. We found that 18% of all families reported not experiencing a negative impact from the childhood cancer diagnosis, whereas 42% reported having a moderate impact and 40% an intense or very intense impact.

Chile is a very centralized country, with most sophisticated health care providers located in the metropolitan region or the most populated municipalities in the northern region of the country (Antofagasta) and in the south (Concepción). Unsurprisingly, 55% of patients were forced to relocate to receive medical treatment, and most of them were born outside the metropolitan region.

### 3.3. Impact on the Patient

The results in Table 3 show that as many as 58% of patients dropped out of school due to the cancer diagnosis, 80% of them had difficulties meeting with friends or carrying out hobbies or sports, and 59% experienced a negative emotional impact. Figure 1 presents a series of emotional impacts, with the most frequent emotional problem being feeling depressed (18%), followed by feeling irritable (16%), and feeling discouraged (12%).

### 3.4. Impact on the Siblings

Table 4 presents results focused on siblings. From the 64 respondents with siblings, 10% reported having some difficulty with school, 24% faced difficulties spending time with others, and 55% experienced a negative impact on their relationship with their parents. Note that one respondent did not answer the follow-up questions on siblings.

### 3.5. Impact on the Primary Caregiver

We summarize the main new activities associated with the cancer diagnosis or treatment for the primary caregivers in Figure 2. Primary caregivers reported taking their child to physician office visits (100%)—most of them accompanied the child during hospitalization (99%) and treatment (97%)—and 67% helped with their child’s rehabilitation.

Other activities not strictly related to medical treatment also had a high frequency. Eighty-six percent of the primary caregivers helped with their child’s education, and 69% had to seek financial assistance to afford the cost of the illness.

These new primary caregivers’ responsibilities may have resulted in difficulties maintaining the same labor market status as before the diagnosis. Accordingly, in Table 5, we observe that 66% of primary caregivers reported having been absent from their jobs, 23% had to quit, 12% changed jobs, and 8% were fired.

In Figure 3, we explore the impact on the caregivers’ relationship with their spouse or partner. From the 75 cases in which the primary caregiver had a partner at the time of the survey, only two reported not having experienced a negative impact from the diagnosis within their relationship. For almost half of the couples, the dimensions “Intimacy” and “Social life” were the most affected, with the highest frequencies and intensity. Around 20% of the couples reported experiencing a strong negative impact on their communication or emotional attachment.

## 4. Regression Analysis

Table 6 provides evidence of the association between the selected impacts of the disease and the characteristics of the patient, the respondent (mostly the primary caregiver) and the family. The probability of reporting that the family of the patient experienced an overall negative impact is associated with the child’s region of birth (95% CI: −0.291 to 0.017). Specifically, it is 13.7 percentage points lower if the child was born in the metropolitan region. This result implies a difference of 10% in the report of a negative impact between patients born in the metropolitan region and other regions of Chile, based on the unconditional probability of 0.816.

Furthermore, the probability of reporting that the patient experienced a negative emotional impact from the disease is positively associated with their age (95% CI: −0.001 to 0.047). Specifically, the probability of reporting that the patient experienced a negative emotional impact from the disease increases 2.3 percentage points with each additional year of age. This result implies an increase of 4%, considering the unconditional probability (equal to 0.58).

The probability of reporting a negative impact on the relationship of the siblings with the parents is 20.8 percentage points lower if the child belongs to a low-SES family (95% CI: −0.427 to 0.012). This result implies a difference of 38.5% in such a probability between patients with low and high SES, based on the unconditional probability of 0.54. Additionally, the probability of reporting a negative impact on the relationship of the siblings with the parents is 21.8 percentage points higher if the child’s primary caregiver is single (95% CI: −0.021 to 0.458). This result implies a difference of 40% in such a probability between patients whose primary caregiver is single and patients whose primary caregiver has a partner.

We found no evidence of an association between the negative emotional impacts of the cancer diagnosis on the primary caregivers’ spouse/partner relationship and the covariates analyzed.

### Relationship between the Negative Impact of the Disease on the Patients, Siblings, Primary Caregiver, and Families

In Figure 4, we explore how outcomes are related to one another. We observed a higher prevalence of a negative impact on the primary caregiver, followed by the impact on the patient, and then the impact on families, followed by the impact on the patient’s siblings. Additionally, social and emotional impacts had a higher incidence than those related to educational and occupational outcomes. Finally, low-SES families had a relatively higher-frequency impact on patients and their families, and a lower-frequency impact on their siblings and caregivers.

## 5. Discussion

This study highlights the psychosocial burden that a blood-related cancer diagnosis in children can have on the lives of patients and their families. We focus on psychosocial costs, mainly on psychological and social well-being, and we contribute to the literature by analyzing this impact at several levels: the family, the child with the cancer diagnosis, their siblings, and the caregiver. The impact on any of these dimensions has the potential to interact and compound, further affecting relationships and dynamics within the household [24]. Most evidence on psychosocial costs focuses on the patient [8], or on the impact on parents [15,25,26]. We aim to provide a more general overview of the effect on families.

The financial burden of pediatric cancer care can be substantial [27]. Borrescio-Higa and Valdes (2021) show that the medical costs of pediatric cancer care are high in Chile, double the per-capita gross domestic product (GDP), with out-of-pocket spending representing almost a third of the annual income of an average policyholder [5]. In this paper, we find that economic fragility increases following a diagnosis of childhood cancer, as many caregivers report job loss and absenteeism. The literature shows that financial difficulties can be a source of psychological distress in the general population [28], including depressive symptoms in parents of children with cancer [26].

However, medical costs are not the only burden on patients and their families. Psychosocial costs are intangible, not precisely defined, and are less documented than economic costs, even though they represent a large and significant part of the total burden of illness [8].

Research shows that caring for a child with cancer is associated with parental stress, post-traumatic stress, and depressive symptoms [12,13,15,26]. Mothers were the primary caregiver in the vast majority of cases in our sample, which is consistent with results from other countries [10,15]. We found that families reported a negative overall impact on the family, and that relationships within the household between the patient and their parents, between siblings and parents and within the caregiver’s spouse/partner relationship were negatively affected [29]. Furthermore, we found that the occupational conditions of the primary caregivers were negatively affected, which is in accordance with the results of a systematic review [27].

Patients reported a range of negative feelings, such as depression, discouragement, and irritability, as well as difficulty in their social lives. These findings are similar to those from other settings [10,16,30,31,32].

One key finding from this study is that moving plays a major role in the disruption of family dynamics, as over half of the families in the sample were forced to move to a new city as a consequence of the diagnosis or treatment. We found that 66% of families of patients born in regions other than the metropolitan region moved to another city following diagnosis. Note that the country capital is located in the metropolitan region, and that 40% of the country’s population lives in this region [23]. This finding points to a barrier in access for those living in other regions of Chile, as the main pediatric cancer care hospital facilities are in the metropolitan region and in the city of Concepcion, the two main destinations of these moves. This finding is in line with results from other Latin American countries, showing barriers in access to diagnostic and treatment centers [33].

Moving can cause disruptions in the routines of siblings, who may also lose parental attention as it shifts to the child with cancer, due to parents devoting more time to caring for the sick child, medical visits, treatment, and hospitalizations. Thus, siblings of children with cancer experience distress and post-traumatic stress symptoms [18,19]. In our sample, we found that over half of the siblings reported a negative impact on their relationship with their parents.

Another interesting result was that SES was not a significant predictor of the overall negative impact on families. Although, as mentioned above, the financial burden of medical care is significant and may affect more economically fragile families, the psychosocial experience of families did not seem to differ across SES. This finding is somewhat different from evidence in Mexico, which shows a positive association between resilience and economic status, although other mediating factors can predict resilience, such as religion and education [34].

Childhood cancer can have a long-lasting impact on survivors, as individuals may experience physical, social, and mental stress post-treatment when they are young adults [35,36]. Our findings are important as the psychological well-being of families is associated with resilience, or the ability to adapt and function when facing adversity [34]. Promoting the creation of interventions that can help families cope with financial stress and promote emotional health and well-being is important. In particular, it is key to promote the early screening of parents and other family members, as it has been shown that higher psychosocial risk at diagnosis is predictive of parental distress at 6 months [10]. Early screening can be linked to interventions that promote resilience in stress management, which have been shown to improve parent-reported resilience and benefit finding [37]. Finally, it is key to decentralize diagnostic facilities, and take measures to facilitate both the logistics and costs of transportation of patients and caregivers [33].

### Limitations

One limitation of this analysis is the cross-sectional nature of the data, which limits the ability to analyze dynamics. Another limitation is the possibility of sampling bias and participation bias with the phone survey, as the final sample is not representative of the population of children with cancer in Chile, particularly in terms of the insurance distribution. Therefore, the results of this study should be interpreted with caution and should be confirmed and complemented by the results of a larger study population. However, due to the fact that the clinical scenarios for children with a diagnosis of cancer are largely independent of insurance type, our main findings are relevant in a broader setting [38]. It is noteworthy that the sample size imposes a limitation on the power to detect statistically significant results. Furthermore, diagnoses were self-reported, and we were not able to confirm the diagnosis with a health professional. Finally, we do not have information on any psychological interventions or external support that may have been provided to families in the course of the child’s treatment. Future research in this setting should be based on longitudinal data, in order to focus on the dynamics and persistence of psychosocial distress and well-being in relation to the time since diagnosis in this context, and compare this to evidence from other countries [25,39].

## 6. Conclusions

Our results highlight the broad impact on the families of children with blood-related cancer following diagnosis, in a country in which this topic has not been previously studied. Most families reported a negative impact upon diagnosis, which was related to disruptions in family dynamics, depressive symptoms in the patient, a poor relationship between the patient’s siblings and their parents, deterioration within the caregiver’s spouse/partner relationship, as well as a worsening of the primary caregiver’s economic condition. Promoting the creation of interventions that can help patients, siblings, and parents cope with distress and promote emotional health and well-being, and developing policies that minimize barriers to access and the need to move across regions for medical treatment, is important.

## Figures and Tables

**Figure 1 ijerph-19-00599-f001:**
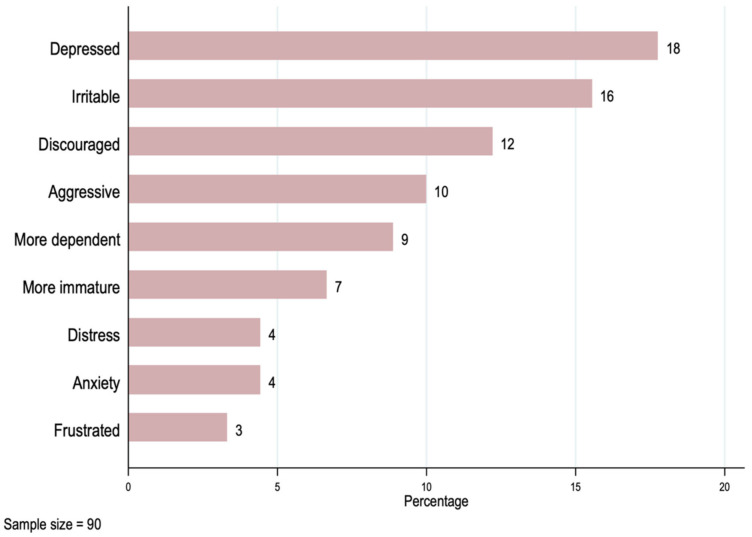
Negative emotional impact on the patient.

**Figure 2 ijerph-19-00599-f002:**
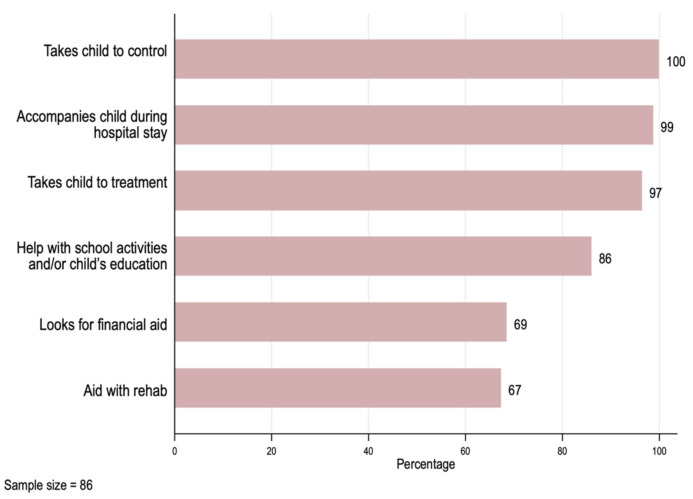
Primary caregivers’ activities related to the patients’ treatment.

**Figure 3 ijerph-19-00599-f003:**
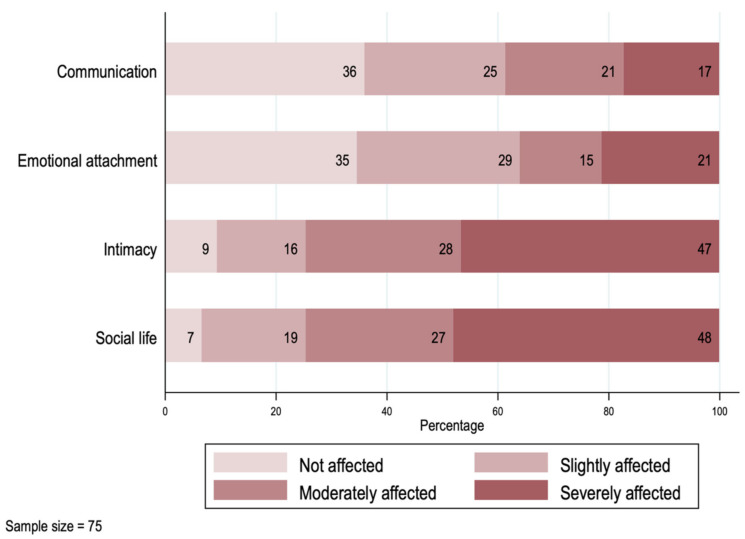
Impact on the primary caregivers’ spouse/partner relationships.

**Figure 4 ijerph-19-00599-f004:**
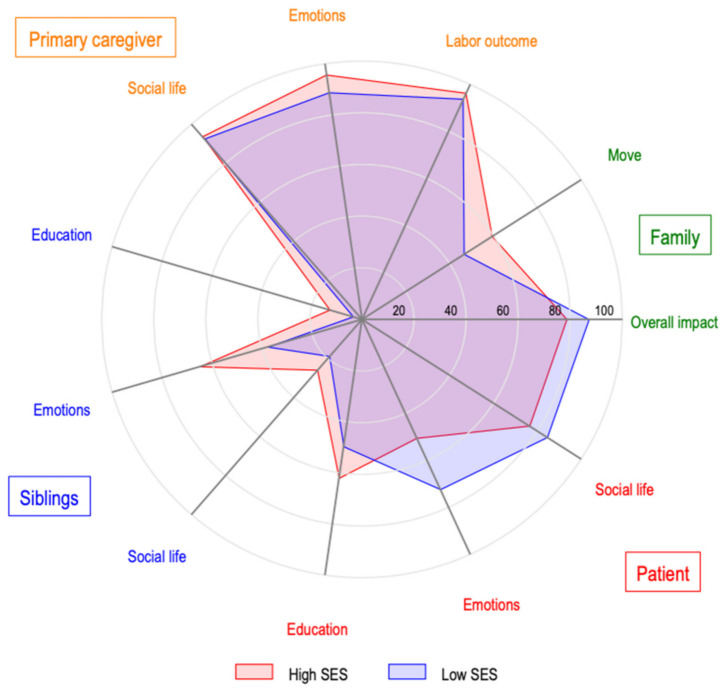
Relationship between the negative impact of the disease on the patients, siblings, primary caregiver, and families.

**Table 1 ijerph-19-00599-t001:** Descriptive statistics.

Variable	Number	Percent	Variable	Number	Percent
*Child’s information*			*Caregivers’ information*		
Gender			Primary caregiver		
Female	47	52.20%	Mother	82	92.10%
Male	43	47.80%	Father	3	3.40%
Age at diagnosis (years)			Mother and Father	2	2.20%
Mean (range)	6.3 (0–16)		Other	2	2.20%
≤2	20	22.20%	Missing	1	1.10%
3–5	23	25.60%	Primary caregiver occupation		
6–8	22	24.40%	Housewife	30	34.50%
≥9	25	27.80%	Unemployed	6	6.90%
Region of birth ^(a)^			Employee—full time	22	25.30%
Metropolitan region	38	43.20%	Employee—part time	16	18.40%
Other regions	50	56.80%	Self-employed	6	6.90%
Diagnosis			Retired	1	1.10%
Leukemia	83	92%	Student	1	1.10%
Non-Hodgkin Lymphoma	7	8%	Other	5	5.70%
Child’s status			Missing	3	3.30%
In treatment	60	66.70%	Is (…) one of the child’s caregivers?		
Treatment ended	28	31.10%	Mother	88	98.90%
Death	2	2.20%	Father	60	67.40%
Health Insurance			Grandmother	42	47.20%
Private	43	47.80%	Sister	15	16.90%
Public	43	47.80%	Aunt	12	13.50%
Other	4	4.40%	Brother	10	11.20%
*Household information*			Grandfather	9	10.10%
Composition			Uncle	2	2.20%
Both parents	44	48.90%			
Single parent	14	15.60%			
Both parents and other adults	9	10.00%			
Single parent and other adults	21	23.30%			
Other adults, no parents	2	2.20%			
Siblings					
Has siblings	65	72.20%			
No siblings	25	27.80%			
Socio-economic status					
Low	33	36.70%			
High	57	63.30%			

Note: ^(a)^ Missing if child had died at the time of the survey.

**Table 2 ijerph-19-00599-t002:** Negative impact on the patients’ families.

Variable	Number	Percent
Number of observations	89	
Negative impact on the family:		
No effect	16	18%
Moderate	37	42%
Intense	28	31%
Very intense	8	9%
Family had to move because of diagnosis and/or treatment	49	55%
Patient born in the metropolitan region ^(a)^	14	29%
Patient born in other region ^(a)^	33	67%
Unknown region of residence ^(a) (b)^	2	4%

Note: ^(a)^ Percentage of families who moved. ^(b)^ Missing if child had died at the time of the survey.

**Table 3 ijerph-19-00599-t003:** Educational, social, and emotional impacts on the patient.

Variable	Number	Percent
Educational impact		
Number of observations	66	
Dropout from school	38	58%
Number of observations	89	
Less time/more difficulties meeting with friends	61	69%
Less time/more difficulties for hobbies or sports	10	11%
Number of observations	90	
Negative emotional impact	53	59%

**Table 4 ijerph-19-00599-t004:** Educational, social, and emotional impacts on the patients’ siblings.

Variable	Number	Percent
Number of observations	64	
*Educational impact*		
Changed school	3	5%
School absenteeism	3	5%
*Social impact*		
Spends more time alone	8	13%
More difficulties meeting with friends	7	11%
*Emotional impact*		
Negative impact on relationship with parents	35	55%

**Table 5 ijerph-19-00599-t005:** Impact on the primary caregivers’ labor market outcomes.

Variable	Number	Percent
Number of observations	86	
Job absenteeism	57	66%
Fired	7	8%
Quit job	20	23%
Change job	10	12%

**Table 6 ijerph-19-00599-t006:** Factors associated with selected impacts of the disease.

Impact		Number of Observations	Unconditional Probability	Marginal Effect	*p*-Value	95% Confidence Interval
Negative impact on the family		87	0.816			
	*Covariate*					
	Child is female			−0.115	0.166	(−0.277; 0.047)
	Child’s age			−0.009	0.360	(−0.03; 0.011)
	Child was born in RM			−0.137	0.082	(−0.291; 0.017)
	Single parent			−0.086	0.286	(−0.244; 0.072)
	Child has siblings			0.089	0.311	(−0.084; 0.262)
	Low SES			0.089	0.343	(−0.095; 0.272)
Negative emotional impact on the patient		88	0.580			
	*Covariate*					
	Child is female			−0.083	0.417	(−0.282; 0.117)
	Child’s age			0.023	0.060	(−0.001; 0.047)
	Child was born in RM			−0.077	0.451	(−0.277; 0.123)
	Single parent			0.061	0.554	(−0.142; 0.265)
	Child has siblings			−0.052	0.656	(−0.28; 0.176)
	Low SES			0.166	0.116	(−0.041; 0.372)
Negative impact on the relationship between siblings and their parents		63	0.540			
	*Covariate*					
	Child is female			0.109	0.348	(−0.118; 0.336)
	Child’s age			−0.019	0.162	(−0.045; 0.007)
	Child was born in RM			−0.039	0.752	(−0.279; 0.201)
	Single parent			0.218	0.074	(−0.021; 0.458)
	Low SES			−0.208	0.064	(−0.427; 0.012)
Negative impact within the primary caregiver’s spouse/partner relationship		73	0.9315			
	*Covariate*					
	Child is female			0.023	0.715	(−0.1; 0.146)
	Child’s age			−0.009	0.247	(−0.025; 0.006)
	Child was born in RM			0.017	0.794	(−0.108; 0.141)
	Single parent			0.097	0.260	(−0.072; 0.266)
	Child has siblings			0.119	0.160	(−0.047; 0.284)
	Low SES			−0.078	0.272	(−0.216; 0.061)

Note: Unconditional probabilities were computed as the number of observations that reported having the specific impact, divided by the total number of observations considered in the regression. Marginal effects were computed using maximum likelihood estimates of the parameters of logit models. Standard errors used to calculate *p*-values are robust to the presence of heteroskedasticity.

## Data Availability

The data presented in this study are available on request from the corresponding author.

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
