# Peer review of "The Psychosocial Burden of Families with Childhood Blood Cancer"

_ijerph, 2022, doi:10.3390/ijerph19010599_

Round 1
Reviewer 1 Report
Cancer is the second leading cause of death for children, and leukemias are the most common pediatric cancer diagnoses in Chile. This report describes the distress, pain, and other negative experiences for patients and their families, as a major part of the overall burden of childhood cancer. The study examines psychosocial experiences in a sample of 90 families of children with blood-related cancer in Chile and provides a global view of the family experience. Not surprisingly, most families report negative experiences, disruptions in family dynamics, and a range of negative emotions in patients, parents and siblings, as well as worsening economic conditions from having to relocate to be close to hospitals. Overall, the paper is an important contribution to the literature on the psychosocial aspects of childhood cancer and it will hopefully lead to greater flexibility and more imaginative support for patients and families on the part of community services and hospitals.
Author Response
We are grateful for all the comments that the reviewers have provided. We consider that the revised manuscript is a better paper than the original manuscript, and we thank the reviewers for pointing us toward it.
Following the suggestions of the reviewers, the current version of the manuscript includes two main changes:
- We have expanded the Introduction and Discussion sections
- We have addressed the methodological concerns and reorganized the results
We have also edited the manuscript for clarity and corrected typos.
Reviewer 2 Report
This is an important area for investigation and one that warrants more thorough discussion. The paper seeks to address the wider effects on families associated with diagnosis of cancer with children and the magnitude of these effects.
The limitations of the approach are highlighted but it would be useful to have a fuller discussion around the methodological concerns and how these might compromise the findings. Further, there needs to be an assessment of the determination of the sample and how this might also bias the findings. While the setting is in Chile, there are likely to be important lessons for other jurisdictions, but some contextual underpinning would prove to be beneficial.
There are some typographical errors that need correcting. For example, the narrative preceding Table 2 indicates that the percentage of families reporting a moderate negative impact was 24% (line 140), while the Table provides a percentage of 42%. Another example is in line 32, where there are words missing.
The paper should be viewed as a preliminary study, which proposes suggestions for further investigation and how greater applicability can be factored into the findings and discussion.
Reviewer 3 Report
This article begings with an important topic of treatment of children with cancer in Chile.
The description of the data is not clearly presented. For example,in lines 93-98, you defined SES as high if the child is a beneficiary of the three applies. However, private health insurance is prevalent among high-income households, when public insurance is divided into four tiers in Chile. Additionally, in Lines 132-134, why is different between the mother of the child’s caregivers and the father of the child's caregivers?
In Section 2.2. Statistical analysis, you should offer more details about the statistical or empirical methods.
In Section 3.2-3.5, the results are the statistical analysis results , I think they would be worthy to be modified as another section.
In Table 6, only a few of the p-value of variables is singficant, your data should be increased and the statistical method shoud be addressed.c
Reviewer 4 Report
Dear Authors,
It is very interesting and important topic but I have some comments:
-
Double [8] for this same paragraph - line 37 and then 41.
-
The period should follow the parenthesis (after reference) for example [8]. not .[8] - it is a problem through the whole text.
- Deeper literature review should be provided. It is difficult to figure out about some other research on this topic - for example research [12] , [13], [14]; what about research on this topic for other countries; ?
- The scale of the problem should be also underline - for Chile but also whether this cancer is also problem of other countries -
- Methodology - more information on the representativeness of this sample should be provided ; the research covers 90 survey responses, then what is the percentage of total children which have one the mentioned diseases? etc;
- Some more information on Chile should be presented to figure out what does mean Metropolitan Region - what kind of area , big etc importance - why this Region was analyzed etc. to what kind of division in Chile it belongs. Why have you chosen this region?
-
"Promoting the creation of interventions that can help families cope with the financial stress and promote emotional health and well-being is important" - lines 287-288 - it would be good to specify what kind of intervention can be undertaken. if you made such detailed research then it would imply to indicate the detailed proposition.
8. "Future research should focus on the dynamic aspect, and the persistence of psychosocial distress and well-being, in relation to time since diagnosis in this context.[21,33]" lines 299-300 - why have you provided references at the end of your conclusion. If such research exist [21,33] then what is the reason to write about future directions of your research ?
9. Discussion part should include more relations with the results of research from other countries and it should be underlined.
Round 2
Reviewer 2 Report
There have been some improvements - documented in the accompanying letter - but a few minor grammatical gliches remain.
Author Response
Thank you very much for your comments.
Following your suggestion, the current version of the manuscript includes one main change:
We have worked with the language editing service at MDPI to correct the grammar, and to improve the language and style of the manuscript.
Reviewer 3 Report
There have been some improvements - documented in the accompanying letter.
Author Response
Thank you very much for your comments.
Following the suggestions of the one reviewer and the Editor, the current version of the manuscript includes one main change:
We have worked with the language editing service at MDPI to correct the grammar, and to improve the language and style of the manuscript.
Reviewer 4 Report
All comments were taken into account and they were answered.
Author Response
Thank you very much for your comments.
Following the suggestions of one reviewer and the Editor, the current version of the manuscript includes one main change:
We have worked with the language editing service at MDPI to correct the grammar, and to improve the language and style of the manuscript.